# Single-Center 20-Year Experience in Surgical Treatment of Malignant Pleural Mesothelioma

**DOI:** 10.3390/jcm11154537

**Published:** 2022-08-03

**Authors:** Giuseppe Mangiameli, Edoardo Bottoni, Umberto Cariboni, Giorgio Maria Ferraroli, Emanuela Morenghi, Veronica Maria Giudici, Emanuele Voulaz, Marco Alloisio, Alberto Testori

**Affiliations:** 1Division of Thoracic Surgery, IRCCS Humanitas Research Hospital, Via Manzoni 56, 20089 Rozzano, Italy; edoardo.bottoni@humanitas.it (E.B.); umberto.cariboni@humanitas.it (U.C.); giorgio_maria.ferraroli@humanitas.it (G.M.F.); veronica.giudici@humanitas.it (V.M.G.); emanuele.voulaz@humanitas.it (E.V.); marco.alloisio@humanitas.it (M.A.); alberto.testori@humanitas.it (A.T.); 2Department of Biomedical Sciences, Humanitas University, Via Rita Levi Montalcini 4, 20090 Pieve Emanuele, Italy; 3Biostatistic Unit, IRCCS Humanitas Research Hospital, Via Manzoni 56, 20089 Rozzano, Italy; emanuela.morenghi@humanitas.it

**Keywords:** malignant pleural mesothelioma (MPM), extended pleurectomy and decortication (eP/D), extrapleural pneumonectomy (EPP), pleura, mesothelioma, thoracic cancer

## Abstract

Objectives: We examined a series of malignant pleural mesothelioma (MPM) patients who consecutively underwent surgery in our institution during the last 20 years. Across this period, we changed our surgical approach to MPM, adopting extended pleurectomy and decortication (eP/D) instead of extrapleural pneumonectomy (EPP). In this study, we compare the perioperative outcomes and long-term survival of patients who underwent EPP vs. eP/D. Methods: A retrospective analysis was carried out of all the MPM patients identified from our departmental database who underwent EPP or P/D from 2000 to 2021. Clavien–Dindo criteria was adopted to score postoperative complications, while Kaplan–Meier methods and a Cox multivariable analysis were used to perform the survival analysis. Results: Of 163 patients, 78 (48%) underwent EPP and 85 (52%) eP/D. Induction chemotherapy was significantly administrated more often in the eP/D group (88% vs. 51%). Complete trimodality treatment including induction chemotherapy, radical surgery, and adjuvant radiotherapy was administered in 74% of the eP/D group versus 32% of the EPP group (*p* < 0.001). The postoperative morbidity rate was higher in the eP/D group (54%) compared to the EPP group (36%) (*p* = 0.02); no statistically significant differences were identified concerning major complications (EPP 43% vs. eP/D 24%, *p* = 0.08). No statistical differences were identified in 30-day mortality, 90-day mortality, median disease-free, and overall survival statistics between the two groups. The Cox multivariable analysis confirmed no induction chemotherapy (HR, 0.5; *p* = 0.002), RDW (HR, 1.08; *p* = 0.02), and the presence of pathological nodal disease (HR, 1.99; *p* = 0.001) as factors associated with worse survival in the entire series. Conclusions: Our data support that eP/D is a well-tolerated procedure allowing the implementation of a trimodality strategy (induction chemotherapy, surgery, and radiotherapy) in most MPM patients. When eP/D is offered in this setting, the oncological results are comparable to EPP. To obtain the best oncological results, the goal of surgical resection should be macroscopic complete resection (R0) in carefully selected patients (clinical N0).

## 1. Introduction

Malignant pleural mesothelioma (MPM) is a rare, aggressive tumor primarily due to excessive exposure to asbestos fiber, which arises from the mesothelial pleural cells [1].

MPM is associated with extremely low survival (5-year survival < 10%), with an average life expectancy of about 12 months [2]. Even now, MPM treatment represents a challenge across the world. Currently, only a few parameters (the stage of the disease, histology, and the patient’s performance status) are usually considered to choose a strategy of treatment, and they appear clearly inadequate if compared with the complexity of this tumor [3].

In nonsurgical patients, systemic chemotherapy is the treatment of choice and is capable of increasing survival by a few months [4]. The role of surgery in MPM is much-debated, even if surgery, when applicable, is probably the only treatment capable of achieving tumor clearance. In patients affected by MPM that is considered to be resectable, two major surgical procedures are currently available to achieve optimal cytoreductive resection: extended (e) pleurectomy and decortication (P/D) and extrapleural pneumonectomy (EPP). However, the presence of unsurmountable anatomical limits and MPM’s diffuse nature make the goal of R0 surgery virtually unattainable. The execution of a very extensive resection procedure, such as extrapleural pneumonectomy (EPP), has been considered the best way to achieve a macroscopic complete resection for a long time [5]. Recently, several retrospective studies and meta-analyses have shown similar or lower long-term survivals and higher perioperative mortalities and postoperative morbidities in patients who have been treated with EPP compared to (e)P/D [6,7]. As a consequence of this evidence, as well as of the results of the MARS Trial, several surgical teams with large experience in EPP have begun to consider (e)P/D as the preferred surgical procedure whenever possible [8]. Across a 20-year period of surgical activity in MPP treatment in our hospital, we have embraced this emerging scientific evidence shifting progressively from EPP (78 procedures between 2000 and 2012) to eP/D (85 procedures since 2010).

In the current study, we analyze our surgical outcomes over a 20-year period (2000–2020) to compare the postoperative morbidity and mortality, the pattern of recurrence, and the survival after eP/D and EPP. We also try to identify MPM patients who could potentially achieve the best oncological advantage from surgical treatment.

## 2. Methods

### 2.1. Patient Selection

A retrospective analysis was performed of all the consecutive patients affected by MPM who were referred to our institution and submitted for radical surgery from 2000 to April 2021. All the patients included in this study were older than 18 years of age and were submitted for cytoreductive surgery through EPP or eP/D after a confirmed histological diagnosis of MPM. No patients who underwent a palliative procedure were included in this study. Pathological staging was defined using the *American Joint Committee on Cancer*, seventh edition. The MPM diagnoses were histologically classified as epithelioid, biphasic, or sarcomatoid according to the final pathological specimens.

All patients signed an informed consent for the acquisition and the usage of clinical data for research purposes at admission. For the use of data, we followed the rules of the Helsinki declaration. The study was approved by the internal research board of our center (Institute Humanitas Clinical and Research Hospital).

### 2.2. Treatment Strategies

In two recent studies we have already described our treatment strategy [9,10]. The general performance status and the cardiopulmonary reserve were systematically evaluated during pretreatment assessment. Contrast-enhanced computer tomography (CT) of the chest and abdomen, as well as FDG-PET scans, were performed for staging all patients. When there was a suspicion of the involvement of extrapleural structures (e.g., the heart) magnetic resonance imaging was also performed [11,12]. As reported, all patients received a histological diagnosis before treatment, and it was obtained through video-assisted thoracoscopic surgery or CT scan biopsy. All cases were discussed in our thoracic malignancies multidisciplinary team meetings. The selection criteria for MPM surgery were: WHO performance status 0–1, FEV1 > 1 L or > 40%, left ventricular ejection fraction >45%, clinical stage I–III (cT3N0), and epithelioid histology. The propensity to administer induction chemotherapy has changed over time, and it was rarely administered before 2007. On the contrary, since 2007 a platinum-based regimen (cis-platinum 75 mg/m^2^ or carbo-platinum AUC 5) and pemetrexed (PEM) 500 mg/m^2^ × 3/4 cycles have been constantly offered to all potential surgical candidates.

All patients were systematically re-evaluated after induction chemotherapy to estimate the response rate by repeating CT, FDG-PET, and ad hoc additional study (endosonography, MRI, or mediastinoscopy). The exclusion criteria for surgery after induction chemotherapy were a worsening of clinical status or a demonstrated disease progression on radiological re-evaluation. From the beginning to 2008, the EPP was the procedure of choice; during the next two years, it was gradually replaced by eP/D and, finally, abandoned after 2010.

Independent of the surgical approach chosen, our standard surgical access was a posterolateral thoracotomy in the VI intercostal space.

In the case of eP/D, the initial dissection was the same as that for extrapleural pneumonectomy, and the visceral pleura was completely removed from the entire lung surface, including in the fissures down to the pulmonary artery. Portions of lung parenchyma invaded by tumor nodules were resected en bloc either by precision resection or by staplers. The pericardium or the diaphragm were resected in the case of macroscopic involvement. The pericardium was reconstructed using a mesh or bovine pericardium patch. Nonpathological parts of the diaphragm were spared as much as possible to simplify primary reconstruction without a prosthetic graft if technically possible. Otherwise, a Proceed^®^ (Ethicon, Johnson&Johnson, Somerville, NJ, USA) or bovine pericardium patch was used. A hilomediastinal lymphadenectomy was systematically performed. In particular, hilar lymph nodes were fully removed with the EPP specimen or systematically harvested in the case of eP/D. Concerning mediastinal lymph node dissection independently of the type of surgery performed, at least two mediastinal stations were systematically harvested.

Intensity-modulated postoperative radiotherapy (PORT) was performed after radical cytoreductive surgery until 2010, and then it was replaced by volumetric-modulated arc therapy (VMAT) technology. The fields of radiotherapy treatment were the costophrenic sinus, chest-wall incisions, sites of chest drains, thoracoscopy tract, and nodal stations involved by tumor and sparing the lung.

For each patient, the following information was collected: age, sex, preoperative hemoglobin (Hb), red blood cell distribution width (RDW), neutrophils, platelets, lymphocytes, neutrophil-to-lymphocyte (LNR) and platelet-to-lymphocyte ratios (PLR), histological classification, induction, postoperative chemo- or radiotherapy, surgical reports, final pathology data, postoperative complications, date and modality of recurrence, and vital status. Clavien–Dindo classification was used to classify surgical complications [13].

Postoperative follow-up was carried out through 3–6 monthly clinical assessments, chest CT scans, and ad hoc further investigation. Follow-up data were also obtained through contact with families and general practitioners, from hospital charts, and from health registries. The follow-up was closed on 1 July 2021.

### 2.3. Outcomes and Statistical Analysis

The primary outcome of this study was the description of disease-free survival (DFS) and overall survival (OS). OS was measured from the day of the surgery until death from any cause or last contact. Disease-free survival (DFS) was measured from the day of the surgery until the first recurrence. Perioperative mortality was defined as death occurring within 30 days of surgery or during the hospital stay. We further verified perioperative mortality by manually reviewing the patients’ records for death.

Categorical data were described as numbers and percentages; for quantitative data, the median and interquartile range (IQR), median and range, or mean and standard deviation were used as appropriate. Owing to low numbers in each pT and pN subcategory, groups including pT 0-1-2 and pN 0 were built, and pT 3-4 and p N+ were left as categorical variables. A survival analysis was performed using the Kaplan–Meier method, with observation times censored to the date of last contact for patients who were still alive. Survival comparisons between groups were performed using the Cox regression method, including in the multivariate model those parameters reaching values of *p* ≤ 0.1 in the univariate analysis. A significance level of 5% (*p* < 0.05) was assumed for comparisons. Stata software (StataCorp. 2017. Stata Statistical Software: Release 15. StataCorp LLC, College Station, TX, USA) was used for all statistical analyses.

## 3. Results

We identified a total of 163 MPM patients who underwent radical cytoreductive surgery. EPP was performed in 78 patients (47.9%) and eP/D in the remaining 85 patients (52.1%). Exclusively, in one case an R2 incomplete ePD was performed due to superior vena cava invasion. The median follow-up time was 117.9 months [38.4–180.1]. Table 1 shows the demographic and clinical characteristics of the patients.

In both groups, the MPM histological diagnosis was most commonly achieved through thoracoscopy (*p* < 0.44). Epithelioid histology was the exclusive histotype in the EPP group and the most common in the eP/D group (88% of cases).

Neoadjuvant induction chemotherapy was more frequently administered to the eP/D group (88.2% vs. 53.8%, *p* < 0.001). Complete trimodality treatment (induction chemotherapy followed by radical surgery and PORT or VMAT (from 2010)) was performed in 74.1% of patients in the eP/D group compared to 32% of patients in the EPP group (*p* < 0.001). No evidence of residual disease in the final surgical specimen was detected in one (1.2%) patient in the EPP group and in three patients (3.5%) in the eP/D group after induction treatment. Definitive pTNM stages are reported in Table 1 for both groups. Pathological nodal involvement was confirmed in 38.5% of patients in the EPP group and in 37.6% of patients in the eP/D group (*p* < 0.43). Mhe mean operative time was significantly different between the two groups (286 ± 49 min in the EPP group vs. 380 ± 73 min in the eP/D group; *p* < 0.001).

In the EPP group, diaphragm resection was performed in 74 patients (94.9%), and in 40 patients (54.1%), it was associated with pericardial resection. Diaphragm reconstruction was more frequently performed with mesh (*n* = 48, 64.8%), followed by directed suture (*n* = 14, 18.9%), bovine pericardium (*n* = 7, 9.4%), and latissimus dorsi muscle flap (*n* = 5, 6.7%). The replacement of the pericardium was performed using mesh (*n* = 27, 67.5%) or bovine pericardium (*n* = 13, 32.5%).

In the eP/D group, diaphragm resection was performed in 83 patients (97.64%), and it was associated with pericardium resection in 34 patients (40%). The replacement of the pericardium was systematically assured using bovine pericardium, and diaphragm reconstruction was more frequently performed with mesh (39 patients; 46.9%), followed by directed suture (38 patients; 44.7%) or replacement with bovine pericardium (6 patients; 7.2%).

The early postoperative course was uneventful in 50 patients (64.1%) in EPP group and in 39 (45.8%) patients in the eP/D group (*p* < 0.02). In the EPP group, the most common major complication (Clavien–Dindo ≥ 3) was respiratory failure (5/12), while the most common minor complication was atrial fibrillation (10/16). In the eP/D group the most common major complication (Clavien–Dindo ≥ 3) was empyema, a occurring in 4 out of 11 complicated patients. The most common postoperative complication (21/46) and the most common minor complication (Clavien–Dindo < 3) for both was prolonged air leak, which contributed to 19 out of 35 minor complications.

The 30-day and 90-day early mortality rates were similar for the EPP and eP/D groups (1.3% vs. 2.3%, *p* = 1.000; and 5.1% vs. 3.5%, *p* = 0.711; respectively). The mean hospital stay was shorter for the EPP group compared to the eP/D group (11 (3–70) vs. 15 (7–70), *p* < 0.001; see Table 1).

Data concerning follow-up and survival are presented in Table 2. The median follow-up was 28.1 months (range 0.1–191.5) for the EPP group and 16.2 months (range 0.7–125.3) for the ePD group. At that time, 2 patients (2.5%) in the EPP group and 24 (28.23%) in the eP/D group were still alive and disease-free. No patients alive with disease were observed in the EPP group, while 16 (18.82%) were observed in the eP/D group. The deaths were 70 (89.7%) in the EPP group and 45 (52.94%) in the eP/D group. MPM recurrence or progression was the cause of death in 58 patients (74.3%) in the EPP group and in 39 patients (45.88%) in the eP/D group. The pattern of recurrence is shown in Table 2.

The median disease-free survival and overall survival lengths were 14.6 (95% CI 5.9–28.8) and 28.1 (95% CI 12.5–39.7) months for the EPP group and 13.7 months (95% CI 9.01–31.71) and 25.5 months (95% CI 15.07–47.5) for the eP/D group (*p* = ns), respectively. In Table 2, the DFS and OS probabilities (%) for each group are reported at 1, 3, and 5 years (see also Figure 1). In Table 3 and Table 4, all the clinical and biological characteristics investigated in the Cox regression analyses of DFS and OS are reported for the entire cohort of patients.

The execution of induction chemotherapy, pT3-4 pathological status, and pathological lymph node involvement (pN+) were significantly associated with DFS in the univariate and multivariate analyses (Table 3). RDW, operation time, the execution of an induction therapy, a pT3-4, and pN+ were prognostic factors significantly related to overall survival in the univariate analysis. They were all confirmed in the multivariate analysis with the exception of pT3-4 pathological status. Concerning the type of surgery performed, no difference was detected between DFS and OS (EPP vs. eP/D) (Table 4, Figure 2).

## 4. Discussion

In this study we reported the perioperative outcomes and long-term survival of patients who underwent radical surgery for MPM (EPP vs. eP/D) in a high-volume center during a 20-year period. In the early 2000s, our surgical strategy was to systematically adopt EPP in treating MPM. However, the technique of MPM surgical resection has been one of the most-debated topics in thoracic surgery in the last two decades [14], during which various scientific evidence has questioned the benefit of EPP in terms of survival extension, as well as for its relative high morbidity and mortality compared to eP/D [14,15,16]. Thus, after a period of two years in which both procedures were carried out, in September 2010 we definitively adopted eP/D as the procedure of choice in treating MPM.

During a 20-year period, we surgically managed 163 patients affected by MPM. The main characteristics of our surgical cohort are comparable with those of other similar retrospective studies [17,18,19,20,21]. As reported in similar studies, the median age in our series was also significantly higher in patients submitted for eP/D compared with the EPP group (65.2 vs. 59.9 years). This was probably a consequence of the surgical team’s attitude to candidates of eP/D as less-fit patients (older, having higher comorbidities and limited lung function) to avoid the higher morbidity rate of EPP. It is known that pneumonectomy with the removal of the hemidiaphragm and ipsilateral pericardium is responsible for significant hemodynamic and respiratory challenge, causing cardiorespiratory complications specific to the procedure [14].

Concerning the pathological stage in our series, the prevalence of stage III was higher in the EPP group (74.3%) compared to eP/D (45.9%). It is difficult to explain this discrepancy in the percentage of stage III disease, which probably reflected patient selection, inadequate nodal sampling, or the effect of the systematic administration of preoperative induction chemotherapy adopted in recent years. However, these rates are comparable with those reported in the literature, ranging from 44.9% [21] to 85% [20] for EPP and from 46.3% [21] to 79.9% [20] for P/D.

Incomplete R2 resection rate was reported in only one retrospective study comparing EPP vs. P/D. In particular, Batirel et al. reported a 46% rate of R2 resection in a series of 130 patients affected by MPM and submitted for surgery (both EPP and P/D). No survival data concerning R2 resections were available in the series [19]. Incomplete R2 resection rates have been reported in most retrospective studies concerning P/D, ranging from 3 to 6% [15,16,17,21,22,23]; only in two series, it was greater than 30% [19,24]. Bolukbas et al. reported a 35% macroscopically incomplete (R2) resection rate in a large series. In their study, one patient out of 31 died postoperatively. The outcomes in that subgroup were not different from those of patients receiving chemotherapy only [24].

In our series, an uneventful postoperative course was significantly more probable for EPP patients compared to eP/D patients. Nevertheless, concerning the occurrence of major complications (Clavien–Dindo ≥ 3), no differences were detected between the two groups. Our data are comparable to those reported in others retrospective series, where the morbidity rates range from 10% to 20% for EPP and from 5% to 6.4% for eP/D [17,19].

Probably, the increased morbidity rate reported in our eP/D group was due to the effect of prolonged air leak, which accounts for the larger part of minor complications and contributes to increased major complication rates. Because of the existing variability in the use of the terms pleurectomy and decortication, a set of definitions have recently been issued to clarify what is meant exactly by P/D, or lung-sparing radical pleurectomy [25]. It is now commonly agreed upon that the whole parietal mediastinal and diaphragmatic pleura should be removed in P/D as in EPP; however, the extent of the visceral pleurectomy is left to the surgeon’s judgment, and some surgeons do not remove the whole visceral pleura if it is normal. In our patients, the visceral pleura was systematically peeled off the entire lung surface, including the fissures.

Prolonged air leak caused by systematic peeling of the visceral pleura was the most common complication resulting in a longer median hospitalization stay in the eP/D group.

Concerning the occurrence of empyema, it was the most common major complication in the eP/D group, occurring in five patients, with a mesh graft used for diaphragm reconstruction in four out five procedures. Only one patient in the EPP group presented empyema as a major complication, and also in this case, the diaphragm reconstruction was assured with a mesh graft. For this reason, in our surgical strategy, we systematically tried to spare the diaphragm and close it, primarily in order to avoid the risk of graft infection.

Our 30- and 90-day mortality rates were comparable in the two groups (EPP: 1.3% and 5.1% vs. eP/D: 2.3% and 3.5%), as well as comparable to other retrospective series (see Table 5) [17,19,20].

To date, several perioperative approaches have been proposed to improve survival in patients affected by MPM. Among all, multimodality strategies combining chemotherapy, surgery, and radiotherapy seem to have improved survival, with series reporting median survival benefits between 17 and 35 months and 5-year survival rates of 15% to 20% [15,26,27,28,29].

In several multimodality therapy trials with EPP, only 50–62% of eligible patients were able to tolerate the full treatment regimen [30,31,32]. This evidence was confirmed by our recent study in which we compared our initial surgical experience with ePD vs. EPP in treating MPM: only the 31% of our patients submitted for EPP were capable to tolerate a trimodality treatment [9]. The current study confirmed these data, considering that only 32% of patients in the EPP group were able to benefit from a full trimodality course compared to 74% of the patients in the eP/D group [10]. A possible explication of these data is the fact that the clinical impact of eP/D on patient general condition was apparently less severe, allowing most patients to complete trimodality regimens [24]. A similar explanation may be the different rates of patients that were submitted for adjuvant RT in the two groups (EPP: 50% vs. eP/D: 77.6%).

Concerning the survival analysis, the overall and disease-free median survival data reported in our series are comparable with the recent literature data [16,17,18,19,20,21,22,24,33,34,35]. We did find no oncological survival benefit of either surgical approach over the other, confirming the results reported in similar large, retrospective series in which EPP and eP/D have been compared [17,18,19,20,21]. Furthermore, our data support eP/D as a less morbid procedure.

In the present study, the recurrence rate was higher in the EPP group compared to the e/PD group (81% vs. 67%, *p* = 0.046). The different rates of recurrence were probably a consequence of the longer follow-up time in the EPP group. In our experience, local recurrence was lower in the EPP group compared to the eP/D group (2% vs. 20%). Interestingly, concerning eP/D, our results are in contrast with a recent series in which P/D was associated with a larger proportion of local recurrence (68.4%) [25]. Two factors can probably explain this finding: the high percentage of patients submitted for PORT after surgery (74%) and the careful selection of patients (54% of pathological stages I and II and 88.2% epithelioid final histology).

The current study allowed us to identify several prognostic factors of better disease-free and overall survival. The multivariable analysis identified three factors significantly associated with disease-free survival: the occurrence of induction chemotherapy, pT3-4, and pN+ pathological status. The administration of induction chemotherapy, operation time, pN+ pathological status, and RDW were the four prognostic factors having significant impacts on OS.

In our study, both induction chemotherapy and trimodality treatments were more frequently performed in the eP/D group than in the EPP group, reflecting a major change in our strategy.

In 2014, a retrospective study collecting data from nine Italian reference centers for thoracic surgery confirmed that the success of EPP submitted in the context of a multimodality treatment was influenced by some factors, such as female sex, epithelial histology, and the administration of induction chemotherapy [29]. Today, multimodality treatment is considered the standard of care in MPM, with various treatment strategies proposed, including neoadjuvant, intraoperative and adjuvant treatments. Our study confirmed that eP/D offered in a multimodality treatment setting had satisfactory long-term oncological results comparable to EPP, with the main advantage of being a very well-tolerated procedure, allowing for the administration of trimodality regimens in most patients.

In our cohort, RDW (a measure of the variation in erythrocyte volume) was a prognostic factor of OS. This finding is very interesting considering that RDW has been recently identified as a new and simple prognostic factor, both for non-neoplastic and neoplastic diseases. Recently, Petrella et al. reported the value of preoperative RDW as a disease-free survival prognostic factor in resected pN1 lung adenocarcinoma [36]. Our study is the first evidence of the prognostic role of RDW in a surgical series of MPM considering that, to date, only one study reported RDW as a significant predictive factor for MM prognosis [37]. On the other hand, other hematological-derived prognostic factors, such as LNR and PLR, were not associated with better overall and disease-free survival, while longer operative time was a prognostic factor having a negative impact on OS.

In our surgical series, pathological node involvement (pN+) was one of the strongest prognostic factors of DFS and OS, as has previously been reported by others studies [29,32,38]. The relatively lower frequency of patients with N+ status in our cohort (38.5% in the EPP group and 37.6% in the eP/D group) was the consequence of careful staging aimed at avoiding heavy surgery for N+ clinical patients. In our analysis, no survival differences were observed between pN1 and pN2 patients. Our data support the recent modification carried out on the eighth TNM classification in which only two N categories were identified, with both intrapleural and extrapleural nodes now grouped into category N1. This modification was justified by the fact that survival is more affected by the number of nodes involved than by the specific anatomical locations of nodal disease [39].

Our study presents several limitations. The main limitation is the retrospective nature of the study, which covered a long time period (about 20 years), during which unknown confounding variables played a role, as well as treatment evolution and patient selection over time. Furthermore, new variables not included in our analysis are today available to stratify the prognosis of surgical patients. In 2021, the WHO Classification of Tumors of the Pleura and Pericardium introduced new possible variables that were not investigated in our analysis, such as nuclear grading or analysis of the most frequently altered genes in diffuse pleural mesothelioma [40]. Strengths of this study are that all the patients were treated and followed in a single center, all surgical procedures were performed by experienced surgeons in a high-volume center with 30 years experience in treating MPM, and the relatively large sample size.

## 5. Conclusions

In conclusion, our data supported the concept that e/PD was a well-tolerated procedure, allowing the implementation of a trimodality strategy (induction chemotherapy, surgery, and radiotherapy) in most MPM patients. In our experience, eP/D offered in a multimodality treatment setting had oncological results comparable to EPP. To obtain the best oncological results, the goal of surgical resection should be macroscopic complete resection (R0) in carefully selected patients (epithelioid histology and clinical N0).

## Figures and Tables

**Figure 1 jcm-11-04537-f001:**
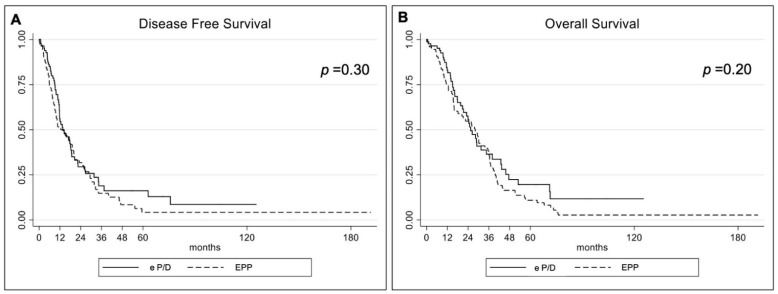
(**A**) Disease-free survival and (**B**) overall survival curves for extrapleural pneumonectomy (EPP) and extended pleurectomy and decortication (e/PD).

**Figure 2 jcm-11-04537-f002:**
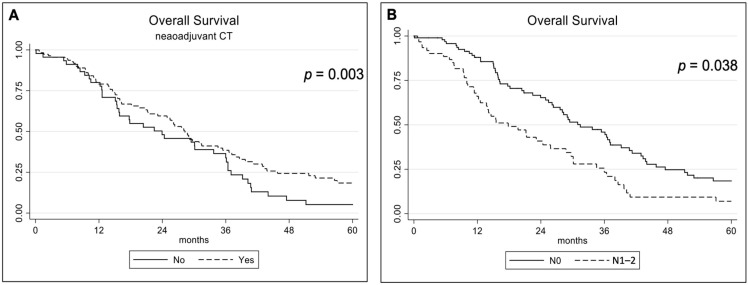
Overall survival curves according to the execution of induction therapy (**A**) and pathological node involvement (pN0 vs. pN+) (**B**).

**Table 1 jcm-11-04537-t001:** Patients’ clinical and pathological features.

Variables	EPP (*n* = 78)	eP/D (*n* = 85)	*p*
Age	59.9 (57.8–61.2)	65.2 (63.5–67.0)	<0.001
Male gender	58 (74.3)	62 (72.9)	0.837
Mean BMI	22.8 (22.2–23.3)	22.7 (22.2–23.3)	0.827
Smoker	29 (37.1)	44 (51.7)	0.137
Asbestos exposure	75/75 (100)	75/80 (93.7)	0.06
Any comorbidity	34 (43.5)	42 (49.4)	0.45
FEV1%	79.4 (75.8–83.0)	79.6 (76.2–82.9)	0.88
RDW	16.6 (15.8–17.4)	16.8 (16.2–17.4)	0.40
NLR%	66.5 (63.6–69.4)	2.98 (2.63–3.34)	0.02
LNF%	21.8 (19.5–24.1)	24.7(23.0–26.5)	0.04
PLR	322.7 (300.5–355.0)	252.7(231.4–273.9)	<0.001
VATS	76 (97.4)	80 (94.1)	0.44
Talc pleurodesis	72 (92.3)	75 (88.2)	0.38
Right-sided tumor	37 (47.4)	49 (57.6)	0.19
Tumor histology			0.088
Epithelioid	78 (100)	75 (88.2)	
Sarcomatoid	–	1 (1.1)	
Biphasic	–	4 (4.7)	
pT category			0.08
T0–1	5 (6.4)	6 (7.0)	
T2	15(19.2)	18 (21.2)	
T3	52 (66.7)	43 (50.6)	
T4	6 (7.7)	18 (21.2)	
pN category			0.43
N0	48 (61.5)	53 (62.3)	
N1	12 (15.4)	18 (21.2)	
N2	18 (23.1)	14 (16.5)	
N+	30 (38.5)	32 (37.6)	
pTNM stage			<0.001
I	4 (5.1)	44 (51.8)	
II	15 (19.2)	2 (2.3)	
III	58 (74.3)	39 (45.9)	
Chemotherapy			
Any induction	42 (53.8)	75 (88.2)	<0.001
Any adjuvant	6 (7.6)	4 (4.8)	0.522
Adjuvant RT	39 (50.0)	66 (77.6)	<0.001
Trimodality therapy	25 (32.0)	63 (74.1)	<0.001
Median hospitalization (days)	11 (3–70)	15 (7–70)	<0.001
Clavien–Dindo	28 (35.9)	46 (54.1)	0.02
<3	16 (20.5)	35 (48.2)	
≥3	12 (15.4)	11 (12.9)	0.088
30-day mortality	1 (1.3)	2 (2.3)	1.000
90-day mortality	4 (5.1)	3 (3.5)	0.711

Values are presented as means with relative 95% CIs, medians with ranges, or numbers and percentages where appropriate. Variables are medians (interquartile range) or numbers (percentages) where appropriate. DLCO%: percent diffusing capacity for carbon monoxide; EPP: extrapleural pneumonectomy; FEV1%: percent forced expiratory volume in 1 s; MCR: macroscopic complete resection; eP/D: extended pleurectomy and decortication; pTNM: pathological TNM.

**Table 2 jcm-11-04537-t002:** Follow-up and survival data.

	EPP (*n* = 78)	Range	eP/D (*n* = 85)	Range	*p*
Median follow-up (months)	28.1	0.1–191.6	16.2	0.7–125.3	0.035
Recurrence (*n*)	59 (80.82%)	%	56 (66.67%)	%	0.046
Local	2	3.39%	11	19.64%	
Local and distant	19	32.20%	35	62.50%	
Distant	0	0%	4	7.14%	
Unknown	38	64.41%	6	10.71%	
Median survival (months)		IQR		IQR	
Disease-free	14.6	5.9–28.8	13.7	9.01–31.71	0.300
Overall	28.1	12.5–39.7	25.5	15.07–47.5	0.207
OS probability (%)	%	95% CI	%	95% CI	
1-year	75.34	63.76–83.69	83.07	72.57–89.82	
3-year	36.99	26.08–47.90	36.48	23.74–49.31	
5-year	10.96	5.12–19.30	19.64	9.27–32–84	
DFS probability (%)	%	95% CI	%	95% CI	
1-year	51.59	39.07–62.74	56.16	44.05–66.63	
3-year	14.80	6.95–25.43	18.93	9.74–30.43	
5-year	4.23	0.81–12.55	16.22	7.58–27.75	

**Table 3 jcm-11-04537-t003:** Details of the statistical analyses: univariate and multivariate analyses for disease-free survival (Cox proportional hazards model).

DFS	Univariable			Multivariable		
Variable	HR	95% (CI)	*p*-Value	HR	95% (CI)	*p*-Value
Age	1.00	0.98–1.02	0.992			
Male sex	1.15	0.75–1.75	0.516			
BMI	1.02	0.95–1.10	0.557			
Any comorbidity	0.75	0.52–1.09	0.134			
Asbestos exposure	1.17	0.54–2.52	0.693			
FEV1%	1.00	0.99–1.01	0.842			
DLCO%	1.01	0.98–1.03	0.689			
RDW	1.00	0.94–1.07	0.885			
NLR	1.04	0.98–1.11	0.184			
PLR	1.01	1.00–1.02	0.223			
Induction therapy	0.53	0.36–0.78	0.001	0.55	0.37–0.81	0.003
Trimodality therapy	0.59	0.41–0.86	0.006			
Operation time	1.12	0.96–1.29	0.144			
Right side	1.06	0.73–1.53	0.766			
Any complication	1.09	0.75–1.58	0.642			
C-D complication ≥ 3	1.26	0.76–2.09	0.367			
pT3-4	2.16	1.38–3.38	0.001	2.01	1.26–3.23	0.004
pN+	1.83	1.25–2.68	0.002	1.54	1.03–2.33	0.038
EPP	1.21	0.84–1.75	0.301			

**Table 4 jcm-11-04537-t004:** Details of the statistical analyses: univariate and multivariate analyses for overall survival (Cox proportional hazards model).

OS	Univariable			Multivariable		
Variable	HR	95% (CI)	*p*-Value	HR	95% (CI)	*p*-Value
Age	1.01	0.99–1.03	0.265			
Male sex	1.12	0.74–1.69	0.586			
BMI	1.02	0.94–1.11	0.588			
Any comorbidity	0.87	0.60–1.25	0.443			
Asbestos exposure	0.85	0.39–1.83	0.675			
FEV1%	1.00	0.99–1.01	0.593			
DLCO%	1.00	0.96–1.03	0.827			
RDW	1.04	0.98–1.11	0.169	1.08	1.01–1.15	0.020
NLR	0.99	0.94–1.05	0.699			
PLR	1.00	0.99–1.01	0.978			
Induction therapy	0.66	0.45–0.97	0.033	0.50	0.32–0.78	0.002
Trimodality therapy	0.71	0.49–1.03	0.073	–		
Operation time	1.15	0.98–1.35	0.097	1.23	1.03–1.48	0.023
Right side	1.08	0.75–1.56	0.671			
Any complication	1.04	0.72–1.50	0.833			
C-D complication ≥ 3	1.58	0.97–2.56	0.064	–		
pT3-4	1.52	1.00–2.30	0.050	–		
pN+	1.98	1.37–2.87	<0.001	1.99	1.34–2.95	0.001
EPP	1.27	0.87–1.85	0.209	–		

**Table 5 jcm-11-04537-t005:** Retrospective comparison of the results of EPP and P/D.

Author	Year	N°	EPP/P/D	Age	Epithelial	Stage 3–4	R^2^	30 D	Median Survival
Flores [17]	2008	663	385/278	60/63 *	269/178	289 (75)/180 (64.7)	-	7/4	12/16
Burt [18]	2014	225	95/130	63/68 *	-	-	-	10.5/3.1	-
Batirel [19]	2016	130	42/66	55.7	97	68	60 (46)	7/2	18.3/14.6
Sharkey [20]	2016	362	133/229	57/65 *	96/173	114 (85)/183 (80)	-	6/3.5	12.9/12.3
Zhou [21]	2021	282	187/95	61/65 *	141/71	84 (44.9)/44 (46.3)	-	7/0	11/18
Current study	2022	163	78/85	60/65 *	78/75	58 (74)/39 (45.8)	0/1	1.3/2.3	28.1/25.5

* *p* < 0.01.

## Data Availability

The data presented in this study are available on request from the corresponding author.

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
