# Peer review of "Single-Center 20-Year Experience in Surgical Treatment of Malignant Pleural Mesothelioma"

_jcm, 2022, doi:10.3390/jcm11154537_

Round 1

Reviewer 1 Report

The article reports what is known about mesothelioma surgery and contains no new information. The necessary information can be obtained in a prospective, multicenter randomized trial that is larger than the study MARS.

The explanation for radiotherapy is insufficient. It is unclear whether all pleural surfaces were irradiated or only the surgical scar of patients who underwent eP/D. If all pleural surfaces were irradiated, a higher rate of radiation pneumonia would be expected.
It is unnecessary to write the information in the tables in detail in the text.
The discussion is too long, and some chapters are redundant.
Some statements have been shown to be incorrect. For example, the p value for tumor location given on page 4 is meaningless. Therefore, the phrase "most common" is not appropriate.
There are some typos throughout the article.

Author Response

Reviewer 1

Comment 1

The article reports what is known about mesothelioma surgery and contains no new information. The necessary information can be obtained in a prospective, multicenter randomized trial that is larger than the study MARS.

The explanation for radiotherapy is insufficient. It is unclear whether all pleural surfaces were irradiated or only the surgical scar of patients who underwent eP/D. If all pleural surfaces were irradiated, a higher rate of radiation pneumonia would be expected.

Answers 1

Thank you for your revision. Now in methods section (2.2. Treatment strategies) we have specified the radiotherapy fields of treatment. In particular the fields of treatment were were the costophrenic sinus, chest wall incisions, sites of chest drains, thoracoscopy tract, and nodal stations involved by tumor, sparing the lung. As reported in the text interestingly, despite the lung was still in place, we did not observe any severe complications of radiotherapy.

Comment 2

It is unnecessary to write the information in the tables in detail in the text. The discussion is too long, and some chapters are redundant.

Answers 2

Thank you for your suggestion. We have now removed several information in the text that can be simply deduce by the tables. We have also reduced the discussion removing redundant sentences.

Comment 3

Some statements have been shown to be incorrect. For example, the p value for tumor location given on page 4 is meaningless. Therefore, the phrase "most common" is not appropriate. There are some typos throughout the article.

Answers 3

Thank you for your suggestion. The sentence concerning tumor localization was effectively incorrect. We have removed the sentences because the relative data can be simply resumed by table. The typos have been correct.

Reviewer 2 Report

This is a nice review to compare the two surgical methods for MPM, (1) EPP and (eP/D). THe results shown in this manuscript showed a very nice prognosis. I think the following points should be added to discussions. 1) With reference to Table 5, the median survival of this report is much better than that of other reports. What does this depend on? What is the bias regarding case selection?  2) Regarding the same Table 5, EPP is better than PD, and isn't it due to the difference in stages after all? In this case group, can we examine the comparison between the two surgical methods for each stage in detail? And, does that give us new insights into the indications such as in what cases PD should be selected?

Author Response

Reviewer 2

Comment 1

This is a nice review to compare the two surgical methods for MPM, EPP and (eP/D). The results shown in this manuscript showed a very nice prognosis. I think the following points should be added to discussions. 1) With reference to Table 5, the median survival of this report is much better than that of other reports. What does this depend on? What is the bias regarding case selection?  

Answers 1

Thank you for your appreciation for our work.

We think that the difference in survival comparing with other series derives from 3 factors:

  1. The low rate of R2 resection compared to other retrospective series. In these series the R” rate ranged from 3% to more than 30%. It was discussed in the discussion:

“Incomplete R2 resection rate has been reported in only one retrospective study comparing EPP vs P/D. In particular, Batirel et al. have reported a 46% of R2 rate resection in a series of 130 affected by MPM and submitted to surgery (both EPP and P/D). No survival data concerning R2 resections are available in the series [19]. Incomplete R2 resection rate has been reported in most retrospective study concerning P/D ranging from 3 to 6% [15-17, 21 -23]; only in two series it was greater than 30% [19, 24]. Bolukbas et al. have reported 35% macroscopically incomplete (R2) resection rate in a large series. In their study, one patient out of 31 died postoperatively. Outcomes in that subgroup were not different from those of patients receiving chemotherapy only [24].”

  1. The selection of treated patients according to histological type. In our series the epithelioid histology rate was very high (88.2%) compared to other series. It was discussed in the discussion in this paragraph:

“In the present study, the recurrence rate was higher in EPP group compared to e/PD group (81% vs 67%, p= 0.046). The different rate of recurrence is probably consequence of longer follow up time in EPP group. In our experience, local recurrence was lower in EPP group compared to eP/D group (2% vs 20%). Interestingly, concerning eP/D, our results are in contrast with a recent series in which PD was associated with a larger proportion of local recurrence (68.4%) [25]. Two factors can probably explain this finding: the high percentage of patients submitted to PORT after surgery (74%) and the careful selection of patients (54% of pathological stage I+II and 88.2% of epithelioid final histology).”

  1. The preoperative selection of patients with all preoperative staging aimed at avoiding heavy surgery for N+ clinical patients. It was discussed in the discussion too:

“ In our surgical series, pathological nodes involvement (pN+) was one of the strongest prognostic factors of DFS and OS as previously reported by others studies [29, 32, 38]. The relatively lower frequency of patients with N+ status in our cohort (38.5% in EPP group and 37.6% in eP/D group) was the consequence of careful staging aimed at avoiding heavy surgery for N+ clinical patients. In our analysis, no survival difference were observed between pN1 and pN2 patients. Our data support the recent modification carried out to the eighth TNM classification in which only two N categories have been identified with both intrapleural and extrapleural nodes now grouped into category N1. This modification was justified by the fact that survival is more affected by the number of nodes involved than by the specific anatomical locations of nodal disease [39].”

Finally all these considerations are now resumed in the conclusion that we have modified as follow:

“To obtain the best oncological results the goal of surgical resection should be macroscopic complete resection (R0) in careful selected patients (epithelioid histology and clinical N0). “

Comment 2

2) Regarding the same Table 5, EPP is better than PD, and isn't it due to the difference in stages after all? In this case group, can we examine the comparison between the two surgical methods for each stage in detail? And, does that give us new insights into the indications such as in what cases PD should be selected?

Answers 2

Thank you very much for this observation. Surely the different rate in pathological stage in the two group influences the final oncological results. Our personal idea is that EPP remains the surgical strategy that more often was offered to higher stages in the first years of 2000 (both in retrospective studied that in our series). Across the time the surgical teams understood that the patients affected by higher stage should not be submitted to surgery but to chemotherapy.  It was confirmed in our series too where, as you can see in the table 1, it was a disproportion in STAGE I between EPP (5.1%, 4 patients) and ePD (51.8%, 44 patients). Due to these unbalanced data we did not believe possible to compare the two surgical strategy for each stage in detail without bias.

Round 2

Reviewer 1 Report

None